REGISTERED REPORT PROTOCOL

# Protocol for the systematic review and meta-analysis of factors associated with non-adherence to antidepressants in depressive disorders in those more than 18 years of age

**Saraswati Dhungana** [1]*, **Rojal Rijal** [2‡], **Binit Regmi** [3‡], **Jala Shree Bajracharya** [4‡], **Subriddhi Sharma** [5‡], **Sunil Singh** [4‡], **Nilam Karn** [6‡], **Manisha Chapagai** [1]

1 Department of Psychiatry and Mental Health, Institute of Medicine, Tribhuvan University, Kathmandu, Nepal, 2 Chitwan Medical College, Bharatpur, Nepal, 3 Nepalese Army Institute of Health Sciences, Kathmandu, Nepal, 4 Weifang Medical University, Weifang, China, 5 College of Medical Sciences, Bharatpur, Nepal, 6 Koshi Hospital, Biratnagar, Nepal

☯ These authors contributed equally to this work.
‡ RR, BR, JSB, SS, SS, and NK also contributed equally to this work.
* iomsaras@gmail.com, saraswati.dhungana@studmed.uio.no

## Abstract

### Objective

We have developed a protocol for the systematic review and meta-analysis of the factors associated with nonadherence to antidepressants in depressive patients more than 18 years of age.

### Methods

We will search articles with the databases PubMed- Medline, Scopus, Embase and PsychINFO. All the published literature reporting factors associated with nonadherence to antidepressants in depressive disorders in patients more than 18 years of age and fulfilling all the eligibility criteria till October 2022 will be included. The data will then be extracted and examined to be included in the systematic review. Finally, we will conduct meta-analysis for factors associated with non-adherence.

### Conclusion

We will do extensive search on the proposed topic within the available literature and come up with a robust review of factors associated with nonadherence to antidepressants in patients age more than 18 years globally. The evidence generated would assist in designing interventions to address non-adherence in this group of patients leading to better productivity and quality of life.

This is a Registered Report and may have an associated publication; please check the article page on the journal site for any related articles.

**Data Availability Statement:** Any pilot data reported in this submission are fully available and data collected during the study will be made fully available without restriction upon study completion. All relevant data are within the paper and its Supporting Information files.

**Funding:** The author(s) received no specific funding for this work.

**Competing interests:** The authors have declared that no competing interests exist.

## Introduction

Depression is amongst the top leading causes of disability globally, estimated to be one of the three leading causes, alongside HIV-AIDS and Ischemic heart disease by 2030 [1, 2]. It causes huge suffering in terms of economy and productivity [1, 2]. It is a chronic condition with high relapse and recurrent rates and increased hospitalization, leading to reduced functioning and poor quality of life and increased healthcare costs [1–4]. Besides clinical and productivity issues, depression is associated with severe direct and indirect costs [1, 4–6]. Renewed evidence from the latest global burden of disease 2017 study calls for immediate action to solve the crisis since this is not just a mental health agenda but a public health problem [1]. Antidepressants are the mainstay of treatment for depressive disorders. Many patients with depressive disorders do not achieve full remission despite taking prescription antidepressants and almost half of these patients have recurrent episodes over time [7]. This requires strict adherence to treatment for patients to be able to function optimally and lead near normal lives [7].

In the last decade, there has been development of a number of safe and effective medications for depressive disorders [8]. However, studies suggest that nonadherence to antidepressants is a big challenge with estimates ranging between 13–55.7% [9] to as high as 50 to 60% [10]. Non- adherence to antidepressants further exacerbates this problem, with resultant increased relapse rates, hospitalization and poor quality of life [2]. Literature suggests a number of reasons for non-adherence, including adverse effects of drugs, nil or minimal response and other patient or doctor related factors [8, 11].

Despite being a huge problem, only scant comprehensive literature is available on non-adherence to antidepressants and factors related. A review and meta-analysis on non-adherence to psychotropics concluded that patients' attitude towards their drugs, insight level, attitude to health and stigma mainly contributed towards nonadherence. Additionally, substance use behavior, sociodemographic variables such as age, gender, education and employment, social factors such as social support and family dynamics, clinical factors such as comorbidities and polypharmacy played roles in non-adherence to psychotropics [10]. We believe that non-adherence to antidepressants is also multifactorial [12] and similar factors account for it as supported by a recent systematic review by Solmi et al. [2, 13]. These factors have been categorized as patient- related and prescriber related [13]. A review by Chong and colleagues on interventions aimed at improving adherence to antidepressants reported that multifaceted interventions involving both service user and provider with behavioral component were effective [1], which was supported by another targeted review [13]. However, a meta-analysis published recently concluded that the interventions are effective only in the first six months [14].

To our knowledge, the only comprehensive literature on factors related to nonadherence to antidepressants in depressive disorders in form of review was published in 2015 [12] and has not been updated to this date. We, therefore, want to conduct a systematic review and meta-analysis to examine the factors associated with non-adherence to antidepressants in those more than 18 years of age.

### Research questions

What are the factors associated with nonadherence to antidepressants in depressive disorders in those more than 18 years?

### Objectives

To identify the factors associated with nonadherence to antidepressants in depressive disorders in those more than 18 years of age and conduct a meta-analysis.

## Materials and methods

### Registration and reporting of the review findings

The systematic review protocol has been registered with the PROSPERO with registration number CRD42021273639. Preferred Reporting Items for Systematic Review and Meta-Analyses: The PRISMA Statement [15, 16] and (PRISMA-P 2015) [17] will be used to report the findings of the review (S1 File).

### Study design

Systematic review and meta-analysis will be used to identify the factors associated with nonadherence to antidepressants in those with depressive disorders in patients more than 18 years of age.

### Eligibility criteria

**Inclusion criteria.** 1. Studies published till October 2022.

2. All quantitative studies from cross- sectional studies to cohort studies to interventional studies and randomized controlled trials.

3. Patients diagnosed as depressive disorder/ clinical depression/ depression/ unipolar depression.

4. Studies that have looked into factors accounting for non-adherence to antidepressants.

5. Those more than 18 years of age.

**Exclusion criteria.** 1. Depressive disorders not under antidepressants.

2. Depressive disorders under psychotherapy of any kind.

3. Bipolar depression.

4. Pediatric depression.

5. Editorials, systematic review, qualitative studies, viewpoint, dissertation, abstracts/presentations, and studies with incomplete data.

6. Studies conducted beyond the publication window.

7. Studies conducted in language other than English.

### PICO/S search criteria

**Population.** Patients more than 18 years of age and diagnosed as depressive disorder/ clinical depression/ depression/ unipolar depression by doctors/ clinicians/ therapists/ psychiatrists and prescribed antidepressants. We will not restrict the search to specific contexts. Individuals with comorbid conditions (e.g., diabetes, cancer, heart disease and substance comorbidities) will be included. The location (e.g., outpatient/inpatient, private practice) and mode of treatment (in person/digital), will also not restrict our search strategies.

**Exposure.** Those patients with depressive disorders and under antidepressants as prescribed by doctors.

**Comparators.** Any comparison groups of the included studies if available will be included in the review. The comparators could be any control groups or comparison groups of the intervention studies or observational studies (e.g., types of antidepressants, age group, single episode versus recurrent depressive disorder).

**Outcome.** Non-adherence to antidepressants.

Non-adherence to antidepressants as prescribed by the treating physicians will be the main outcome. For the operational definition of non-adherence, we will follow the definition given by World Health Organization, which states that non-adherence is "the extent to which a person's behavior- taking medication, following a diet, and/or executing lifestyle changes does not correspond with agreed recommendations from a health care provider" [18]. Non-adherence to antidepressants has the potential to be labelled as dichotomous as subjective responses (Yes/No), categorical (high, moderate, low, very low) or continuous (in terms of scores in use of scales). All studies using any assessment tool will be included in the study.

**Studies (observational and interventional).** Studies conducted globally and available in the search databases within the publication window will be considered.

## Search strategies

Databases for search will be PubMed- Medline, Scopus, Embase and PsychINFO. We will use combination of keywords, related medical subject heading (MeSH) terms and other relevant search terms to screen the articles. The search terms used will be 1) "Factors" OR "determinants" OR "predictors" OR "causes" OR "risk factors"; 2) "Nonadherence" OR "non-adherence" OR "noncompliance" OR non-compliance" OR "adherence" OR "compliance"; 3) "antidepressants" OR "antidepressant drugs" OR "depression medication/s" OR "depression drugs" OR "drugs" OR "medications"; 4) "depressive disorders" OR "depression" OR "unipolar depression" OR "clinical depression". The MeSH terms and keywords in title/abstract will be used in the 1+2+3+4 format in the PubMed- Medline search (S2 File). For other search databases, we will use relevant search terms with Boolean operators as applicable. Searches will be re-run before the final analysis.

**Data extraction (selection and coding).** *Study selection.* Two reviewers (BR and RR) will independently screen and select studies to be included in the first stage using the search strategies by title and abstract based on the inclusion criteria. Discrepancies in the first stage will be resolved by third and fourth reviewers (SD and MC). Then, all the authors will independently review and agree on the full text of all selected studies to assess their suitability in the inclusion process. Zotero will be used to collect, organize, and manage literature and to keep record and manage duplicates.

**Data extraction.** Data will be extracted using an extracting data form designed for the purpose of this review by JS and SuS. All studies retrieved from our search strategies using our PICO/S search questions and inclusion criteria will be imported to Zotero and managed in different folders as necessary. All the factors associated with nonadherence as 1. patient- related factors, 2. social- economic factors, 3. health system factors, 4. condition- related factors and 5. therapy- related factors will be extracted. Below is the list of details to be included for each study included in the final stage after risk of bias assessment. In case of missing data, the authors will be contacted as necessary.

a. Author and year/ date of publication

b. Country and setting

c. Sample size

d.  Participant ages

e.  Participant genders (N, %)

f.  Mean age (SD, range)

g.  Any comorbidity

h.  Antidepressant used (if stated)

i.  Study duration

j.  Research aims

k.  Research design

l.  Measures used to assess antidepressant non-adherence

m.  Factors associated with non-adherence

n.  Summarize results and key message

**Risk of bias (quality) assessment.**   Standardized tools applicable to different studies as listed below will be used to assess risk of bias assessment and 2 reviewers (SS and NK) will take part in the process. Disagreements in the judgement regarding risk of bias assessment will be sorted by further discussion with SD and MC. Risk of bias assessment for randomized controlled trials version 2 (RoB 2) [19] will be used to assess risk of bias for randomized controlled studies, where we will use the tool to assess the risk in five specific areas, namely bias arising from randomization process, deviation from intended interventions, missing outcome data, measurement of outcome, and selection of reported results. ROBINS-I [20] will be used to assess risk of bias in the results of non-randomized studies that compare health effects of two or more interventions. NIH quality assessment tool (Study Quality Assessment Tools | NHLBI, NIH) or other suitable and standard quality assessment tools such as The Newcastle-Ottawa Scale (NOS) [21] will be used for specific designs of the observational studies as appropriate. ROBvis tool [22] will be used to generate traffic light plots. Grades of recommendation, Assessment, Development, and Evaluation (GRADE) [23] process will also be used eventually to check overall quality of the studies included and will be categorized as high, moderate, low and very low using GRADEpro. This will be done by SD and MC.

**Data synthesis strategy and subgroup analysis.**   We plan to bring out a narrative synthesis of the results highlighting the main outcomes of interest. We will provide the measures used to assess non-adherence to antidepressants and also the factors associated with non-adherence by classifying them under specific headings identified as 1. patient- related factors, 2. social-economic factors, 3. health system factors, 4. condition-related factors and 5. therapy-related factors. Findings extracted from studies will be summarized in a summary table. A narrative synthesis of the results will be also provided and organized in topics, according to the specific exposures. The quality of the studies will be also discussed.

Our main outcome of interest is nonadherence to antidepressants. Non-adherence to antidepressants has the potential to be labelled as dichotomous as subjective responses (Yes/ No), or continuous (in terms of scores in use of scales).

For studies with dichotomous outcomes, we plan to report odd ratios, while for those with continuous outcomes, such as mean scores on adherence, we will report means for individual studies first, followed by reporting a combined effect estimate. If we find sufficient studies allowing meta-analysis, we will illustrate with a forest plot to provide an overall summary of

effect estimates using odd ratios and standardized mean differences with confidence intervals for individual studies and overall.

We assume that the studies included will be heterogenous, therefore, we plan to apply random effect model. We will measure the extent of heterogeneity across the studies included using $I^2$ (Inconsistency index) [24]. $I^2$ 50%-90% will be considered to have substantial heterogeneity while 75% to 100% will be considered to have considerable heterogeneity. Variance within studies will be estimated by inverse variance weighting using residual maximum likelihood (REML) technique [25]. We consider doing analysis of subgroups if there are good number of studies to allow subgroup analysis for example, if adequate number of studies are available in terms of specific antidepressants used to see which group of antidepressants are associated with more nonadherence to antidepressants. We also plan to do subgroup analysis based on age groups like 18 to 45, 45 to 60 and 60 and above if this is applicable.

## Conclusion and limitations

Depressive disorders are one of the most disabling mental health conditions [26]. Evidence suggests that adherence to antidepressants is vital in preventing recurrences and lifelong disability [27]. Despite robust evidence of safe and effective drugs in most cases, nonadherence to antidepressants is amongst the big challenges faced in management of depressive disorders [28, 29]. The consequences are increased healthcare costs due to high rates of hospitalization and emergency visits, chronicity and residual symptoms, reduced functioning, disability and poor quality of life [2–5].

It is therefore imperative to understand the factors associated with nonadherence to antidepressants in patients with depression. Knowledge of the factors or determinants of nonadherence from a global perspective will help formulate treatment plans for these patients to improve the overall problem of nonadherence so as to decrease the disability caused by this condition leading to improved productivity and economic prosperity.

This systematic review, however, will not be without limitations. The biggest limitation would be the heterogeneity of study designs given the broad nature of the topic to be reviewed. Further, studies published in language other than English would be excluded and also, studies could be missed since we will not be using all the databases.

## Supporting information

**S1 File. PRISMA guide.**
(DOCX)

**S2 File. Search keywords.**
(DOCX)

## Author Contributions

**Conceptualization:** Saraswati Dhungana, Manisha Chapagai.

**Data curation:** Saraswati Dhungana, Manisha Chapagai.

**Formal analysis:** Rojal Rijal, Binit Regmi.

**Investigation:** Saraswati Dhungana, Rojal Rijal, Jala Shree Bajracharya, Subriddhi Sharma, Manisha Chapagai.

**Methodology:** Binit Regmi, Sunil Singh, Nilam Karn.

**Project administration:** Rojal Rijal, Binit Regmi.

**Software:** Rojal Rijal, Binit Regmi.

**Writing – original draft:** Saraswati Dhungana, Manisha Chapagai.

**Writing – review & editing:** Saraswati Dhungana, Rojal Rijal, Binit Regmi, Jala Shree Bajracharya, Subriddhi Sharma, Sunil Singh, Nilam Karn, Manisha Chapagai.

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
