## [Decision Letter · Decision Letter 0]

21 Nov 2022

PONE-D-21-32408Protocol for the systemic review and meta-analysis of factors associated with non-adherence to antidepressants in depressive disorders in those more than 18 years of agePLOS ONE

Dear,

Thank you for submitting your manuscript to PLOS ONE. After careful consideration, we feel that it has merit but does not fully meet PLOS ONE’s publication criteria as it currently stands. Therefore, we invite you to submit a revised version of the manuscript that addresses the points raised during the review process. Please submit your revised manuscript by 21 December 2022. If you will need more time than this to complete your revisions, please reply to this message or contact the journal office at plosone@plos.org. Please include the following items when submitting your revised manuscript:A rebuttal letter that responds to each point raised by the academic editor and reviewer(s). You should upload this letter as a separate file labeled 'Response to Reviewers'.A marked-up copy of your manuscript that highlights changes made to the original version. You should upload this as a separate file labeled 'Revised Manuscript with Track Changes'.An unmarked version of your revised paper without tracked changes. You should upload this as a separate file labeled 'Manuscript'.

We look forward to receiving your revised manuscript.

Kind regards,

Muhammad Shahzad Aslam, Ph.D.,M.Phil., Pharm-D

Academic Editor

PLOS ONE

Journal Requirements:

2. In your cover letter, please confirm that the research you have described in your manuscript, including participant recruitment, data collection, modification, or processing, has not started and will not start until after your paper has been accepted to the journal (assuming data need to be collected or participants recruited specifically for your study). In order to proceed with your submission, you must provide confirmation.

Reviewers' comments:

Reviewer's Responses to Questions

**Comments to the Author**

1. Does the manuscript provide a valid rationale for the proposed study, with clearly identified and justified research questions?

Reviewer #1: Yes

Reviewer #2: Partly

2. Is the protocol technically sound and planned in a manner that will lead to a meaningful outcome and allow testing the stated hypotheses?

Reviewer #1: Yes

Reviewer #2: Yes

3. Is the methodology feasible and described in sufficient detail to allow the work to be replicable?

Reviewer #1: Yes

Reviewer #2: No

4. Have the authors described where all data underlying the findings will be made available when the study is complete?

Reviewer #1: Yes

Reviewer #2: Yes

5. Is the manuscript presented in an intelligible fashion and written in standard English?

Reviewer #1: Yes

Reviewer #2: Yes

6. Review Comments to the Author

You may also provide optional suggestions and comments to authors that they might find helpful in planning their study.

Reviewer #1: The authors present a protocol for a systematic review and meta-analysis of factors associated with medication non-adherence to antidepressants in individuals with depressive disorder. They propose a clear, well presented and structured protocol.

The authors propose to update this research question with recent data from the literature. To do so, they will draw on previous reviews of the literature on this topic that may be worth citing in the introduction part.

The authors chose to include in their review of the literature, studies including individuals with depressive disorder but also with somatic and psychiatric comorbidities. It might be interesting to justify this choice, which is not common, given that in many studies, serious somatic comorbidities such as cancer are a reason for exclusion. This inclusion criterion could be both a strength and a limitation of this future study.

Finally, the authors propose to classify the factors associated with non-adherence to antidepressants into 5 categories: 1. Patient factors, 2. Community and social factors, 3. Health facility-health provider, 4. Health system, 5. Other factors. However, it might be relevant to use the quite similar categories determined by the WHO in 2003 to list these factors, which are: patient-related factors, social-economic factors, health system factors, condition-related factors and therapy-related factors. In addition, some authors in the literature are increasingly referring to intentional and unintentional adherence to classify the factors associated with non-adherence (Wroe et al., 2002; Lehane et al., 2007).

Reviewer #2: Thank you for the opportunity to review the registered report protocol “Protocol for the systemic review and meta-analysis of factors associated with non-adherence to antidepressants in depressive disorders in those more than 18 years of age”

Overall I believe this protocol is lacking in detail, however is otherwise sound and could be improved if the authors consider the below queries.

The PROSPERO listing needs updating – says anticipated completion 01 October 2022, and hasn’t been updated since February 2022

The largest issue however is the lack of methodological detail for the meta-analysis. As it stands, in my opinion this protocol could not be reproduced based on information provided in this manuscript.

Specific queries as follows:

Abstract:

Please provide more detail in the methods regarding meta-analytic techniques. There is currently no method proposed

Consider commenting on the impact of your review in your conclusion. Why is this important?

Introduction:

Consider re-wording this section to be less emotive and more specific. For example, line 49 states “enormous suffering.” This term is unclear, and the paper would be improved by being more specific in your descriptions of the impact of depression in both the introduction and discussion

Line 61 – remove “to name a few”

Line 64 – what do you mean by updated? I don’t think you have adequately described the background literature and previous systematic reviews, so I’m struggling to see where this sits in the context of the field. A quick search online yielded the below, which could be considered

Chong WW, Aslani P, Chen TF. Effectiveness of interventions to improve antidepressant medication adherence: a systematic review. International journal of clinical practice. 2011 Sep;65(9):954-75.

Solmi M, Miola A, Croatto G, Pigato G, Favaro A, Fornaro M, Berk M, Smith L, Quevedo J, Maes M, Correll CU. How can we improve antidepressant adherence in the management of depression? A targeted review and 10 clinical recommendations. Brazilian Journal of Psychiatry. 2020 Jun 1;43:189-202.

Ho SC, Chong HY, Chaiyakunapruk N, Tangiisuran B, Jacob SA. Clinical and economic impact of non-adherence to antidepressants in major depressive disorder: a systematic review. Journal of affective disorders. 2016 Mar 15;193:1-0.

Lingam R, Scott J. Treatment non‐adherence in affective disorders. Acta Psychiatrica Scandinavica. 2002 Mar;105(3):164-72.

Semahegn A, Torpey K, Manu A, Assefa N, Tesfaye G, Ankomah A. Psychotropic medication non-adherence and its associated factors among patients with major psychiatric disorders: a systematic review and meta-analysis. Systematic reviews. 2020 Dec;9(1):1-8.

Line 64-65 – remove “this very important topic relating to”

Line 67-69 – Why isn’t meta-analysis mentioned in your statement of aims?

Materials and methods

Line 83 - typo "metanalyses" please amend

Line 88 – why is this restricted to July 2021? Studies are already 16 months out of date

Line 133-138 – can be removed, that information was stated above

Line 214-215 – the protocol should pre-dictate the analysis plan regardless of the data to remove bias in the reporting. Why is meta analysis dependent on prior review of the results? Further, what does “This will be checked upon by SS and NK” mean?

More information about the meta-analysis is required. What kind of effects models will be used? How is significant and substantial heterogeneity defined? How will you estimate variance? What sub group analyses are planned (these should be pre-planned to avoid bias)?

Conclusions

Line 232-234 – Please be more specific in the impacts of medication adherence (i.e. remove terms like enormous suffering and add specific sequalae) and reference these statements

7. PLOS authors have the option to publish the peer review history of their article (what does this mean?). If published, this will include your full peer review and any attached files.

Reviewer #1: No

Reviewer #2: No

---

## [Author Response · Author response to Decision Letter 0]

4 Dec 2022

December 4, 2022

Dear Editors and reviewers

PLOS ONE 

We thank you for your sincere and thorough review of our systematic review protocol “Protocol for the systematic review and meta-analysis of factors associated with non-adherence to antidepressants in depressive disorders in those more than 18 years of age” manuscript number PONE-D-21-32408 and providing us the opportunity to revise it. We now better understand the critical issues with the paper after the comments made by both the reviewers. We have reviewed the protocol incorporating all the comments made by the reviewers as sincerely as possible. We have attempted to respond to each query to the best of our knowledge. We have revised our introduction, methods and conclusion section and in doing so, we have added 11 references. We now believe that the quality of the manuscript has greatly improved.

Please find our point by point responses to the comments made by the reviewers as follows, where we have highlighted the comments in blue for easy readability with lines in the tracked original version where changes have been made. We have also submitted the original manuscript with track changes, revised manuscript along with the Response to the reviewers. If there are further queries and comments from you and the reviewers, we would be happy to address them.

Responses to reviewers.

Reviewer #1: The authors present a protocol for a systematic review and meta-analysis of factors associated with medication non-adherence to antidepressants in individuals with depressive disorder. They propose a clear, well presented and structured protocol.

Authors’ response: Thank you for the positive comment. 

The authors propose to update this research question with recent data from the literature. To do so, they will draw on previous reviews of the literature on this topic that may be worth citing in the introduction part.

Authors’ response: Thank you for pointing out this important gap in our manuscript, also raised by the second reviewer. We now have cited important articles to support our research question as follows:

Lines 76-92:

“Despite being a huge problem, only scant comprehensive literature is available on non- adherence to antidepressants and factors related. A review and meta-analysis on non-adherence to psychotropics concluded that patients’ attitude towards their drugs, insight level, attitude to health and stigma mainly contributed towards nonadherence. Additionally, substance use behavior, sociodemographic variables such as age, gender, education and employment, social factors such as social support and family dynamics, clinical factors such as comorbidities and polypharmacy played roles in non-adherence to psychotropics (10). We believe that nonadherence to antidepressants is also multifactorial (12) and similar factors account for it as supported by a recent systematic review by Solmi et al. (2,13). These factors have been categorized as patient- related and prescriber related (13). A review by Chong and colleagues on interventions aimed at improving adherence to antidepressants reported that multifaceted interventions involving both service user and provider with behavioral component were effective (1), which was supported by another targeted review (13). However, a meta-analysis published recently concluded that the interventions are effective only in the first six months (14).”

The authors chose to include in their review of the literature, studies including individuals with depressive disorder but also with somatic and psychiatric comorbidities. It might be interesting to justify this choice, which is not common, given that in many studies, serious somatic comorbidities such as cancer are a reason for exclusion. This inclusion criterion could be both a strength and a limitation of this future study.

Authors’ response: Thank you for the remark. We agree that inclusion of somatic and psychiatric comorbidities is not common in such studies. However, we feel that comorbidities are quite common in depression and by doing a thorough review of articles with comorbidities, we want to investigate the non-adherence pattern in both the groups, with and without comorbidities and if the factors affecting non- adherence are similar or different. We therefore think that this will be a strength to our study and gives direction to interventions to be tailored accordingly.

Finally, the authors propose to classify the factors associated with non-adherence to antidepressants into 5 categories: 1. Patient factors, 2. Community and social factors, 3. Health facility-health provider, 4. Health system, 5. Other factors. However, it might be relevant to use the quite similar categories determined by the WHO in 2003 to list these factors, which are: patient-related factors, social-economic factors, health system factors, condition-related factors and therapy-related factors. In addition, some authors in the literature are increasingly referring to intentional and unintentional adherence to classify the factors associated with non-adherence (Wroe et al., 2002; Lehane et al., 2007).

Authors’ response: Thank you for the comment. We agree that factors determined by WHO in 2003 are similar and more comprehensive compared to our categorization. We therefore plan to categorize the factors accordingly and have changed our categorization of factors to those listed by WHO as follows under section of data synthesis strategy and sub-group analysis.

Lines 202-203 and 245-247 “1. patient- related factors, 2. social- economic factors, 3. health system factors, 4. condition- related factors and 5. therapy- related factors.”

However, for the purpose of this manuscript, we are not considering including the classification of factors as intentional and non- intentional. However, we thank you for letting us know about this categorization in the literature.

Reviewer #2: Thank you for the opportunity to review the registered report protocol “Protocol for the systemic review and meta-analysis of factors associated with non-adherence to antidepressants in depressive disorders in those more than 18 years of age”

Overall I believe this protocol is lacking in detail, however is otherwise sound and could be improved if the authors consider the below queries.

The PROSPERO listing needs updating – says anticipated completion 01 October 2022, and hasn’t been updated since February 2022

The largest issue however is the lack of methodological detail for the meta-analysis. As it stands, in my opinion this protocol could not be reproduced based on information provided in this manuscript.

Authors’ response: Thank you for thorough scrutiny and constructive comments on our manuscript for systematic review protocol. We agree the protocol lacks detail, especially in regard to meta- analytic techniques in methodology. We, therefore, have revised our protocol providing details in the methods section for metanalytic techniques we plan to conduct. 

Thank you for pointing out the PROSPERO issue. We now have updated the protocol in the PROSPERO and have changed the completion date to October 2023 since we are currently in data extraction phase.

Specific queries as follows:

Abstract:

Please provide more detail in the methods regarding meta-analytic techniques. There is currently no method proposed

Consider commenting on the impact of your review in your conclusion. Why is this important?

Authors’ response: Thank you for the comment. We now have provided more detail in the methods as

Lines 42-44 “Finally, we will conduct meta-analysis for factors associated with non-adherence and illustrate the results with a forest plot using Stata or Revman”. We also added 

in lines 48-50 “The evidence generated would assist in designing interventions to address non-adherence in this group of patients leading to better productivity and quality of life.” to our conclusion as you suggested.

Introduction:

Consider re-wording this section to be less emotive and more specific. For example, line 49 states “enormous suffering.” This term is unclear, and the paper would be improved by being more specific in your descriptions of the impact of depression in both the introduction and discussion.

Authors’ response: Thank you for this important comment. We have revised the statement and made it more specific both in introduction and discussion.

In introduction, we added the following.

Lines 52-58: “estimated to be one of the three leading causes, alongside HIV-AIDS and Ischemic heart disease by 2030 (1,2). It causes huge suffering in terms of economy and productivity (1,2). It is a chronic condition with high relapse and recurrent rates and increased hospitalization, leading to reduced functioning and poor quality of life and increased healthcare costs (1-4). Besides clinical and productivity issues, depression is associated with severe direct and indirect costs (1, 4-6).”

Lines 69-92: “with estimates ranging between 13 – 55.7% (9) to as high as 50 to 60% (10). Non- adherence to antidepressants further exacerbates this problem, with resultant increased relapse rates, hospitalization and poor quality of life (2). Literature suggests a number of reasons for non-adherence, including adverse effects of drugs, nil or minimal response and other patient or doctor related factors (8,11). 

In discussion, we added the following.

Lines 285-288: “The consequences are increased healthcare costs due to high rates of hospitalization and emergency visits, chronicity and residual symptoms, reduced functioning, disability and poor quality of life (2-5).”

Line 61 – remove “to name a few”

Authors’ response: We have removed it.

Line 64 – what do you mean by updated? I don’t think you have adequately described the background literature and previous systematic reviews, so I’m struggling to see where this sits in the context of the field. A quick search online yielded the below, which could be considered.

Chong WW, Aslani P, Chen TF. Effectiveness of interventions to improve antidepressant medication adherence: a systematic review. International journal of clinical practice. 2011 Sep;65(9):954-75.

Solmi M, Miola A, Croatto G, Pigato G, Favaro A, Fornaro M, Berk M, Smith L, Quevedo J, Maes M, Correll CU. How can we improve antidepressant adherence in the management of depression? A targeted review and 10 clinical recommendations. Brazilian Journal of Psychiatry. 2020 Jun 1;43:189-202.

Ho SC, Chong HY, Chaiyakunapruk N, Tangiisuran B, Jacob SA. Clinical and economic impact of non-adherence to antidepressants in major depressive disorder: a systematic review. Journal of affective disorders. 2016 Mar 15;193:1-0.

Lingam R, Scott J. Treatment non‐adherence in affective disorders. Acta Psychiatrica Scandinavica. 2002 Mar;105(3):164-72.

Semahegn A, Torpey K, Manu A, Assefa N, Tesfaye G, Ankomah A. Psychotropic medication non-adherence and its associated factors among patients with major psychiatric disorders: a systematic review and meta-analysis. Systematic reviews. 2020 Dec;9(1):1-8.

Authors’ response: Thank you for pointing out this discrepancy and listing out the relevant reviews. We now have included all the articles in the introduction section to give a background on the non- adherence and factors associated relevant to the topic and why this review is needed as follows.

Lines 76-92: “Despite being a huge problem, only scant comprehensive literature is available on non- adherence to antidepressants and factors related. A review and meta-analysis on non-adherence to psychotropics concluded that patients’ attitude towards their drugs, insight level, attitude to health and stigma mainly contributed towards nonadherence. Additionally, substance use behavior, sociodemographic variables such as age, gender, education and employment, social factors such as social support and family dynamics, clinical factors such as comorbidities and polypharmacy played roles in non-adherence to psychotropics (10). We believe that nonadherence to antidepressants is also multifactorial (12) and similar factors account for it as supported by a recent systematic review by Solmi et al. (2,13). These factors have been categorized as patient- related and prescriber related (13). A review by Chong and colleagues on interventions aimed at improving adherence to antidepressants reported that multifaceted interventions involving both service user and provider with behavioral component were effective (1), which was supported by another targeted review (13). However, a meta-analysis published recently concluded that the interventions are effective only in the first six months (14).”

Lines 95-97: “the only comprehensive literature on factors related to nonadherence to antidepressants in depressive disorders in form of review was published in 2015 (12) and has not been updated to this date.” 

Line 64-65 – remove “this very important topic relating to”

Authors’ response: We removed this.

Line 67-69 – Why isn’t meta-analysis mentioned in your statement of aims?

Authors’ response: We revised our aims. Thank you for the comment.

Materials and methods

Line 83 - typo "metanalyses" please amend

Authors’ response: We revised it.

Line 88 – why is this restricted to July 2021? Studies are already 16 months out of date

Authors’ response: Initially we planned to complete the study by October 2022, and this was submitted quite some time back, hence the discrepancy in the date. Now, we plan to include articles till October 2022, which we also revised in our manuscript protocol and complete the study by October 2023. We also changed the expected date of completion in the PROSPERO list registered.

Line 133-138 – can be removed, that information was stated above

Authors’ response: We removed line 133-138.

Line 214-215 – the protocol should pre-dictate the analysis plan regardless of the data to remove bias in the reporting. Why is meta-analysis dependent on prior review of the results? Further, what does “This will be checked upon by SS and NK” mean?

More information about the meta-analysis is required. What kind of effects models will be used? How is significant and substantial heterogeneity defined? How will you estimate variance? What sub group analyses are planned (these should be pre-planned to avoid bias)?

Authors’ response: Thank you very much for pointing out this flaw in our protocol. We agree the analysis plan should be well explained in the protocol prior to starting the study to minimize bias in reporting. We now have explained what we intend to do in our meta- analysis. Initially, we thought that review should be completed before planning to de meta- analysis and based on what our findings look like, we will plan to use meta- analytic techniques as appropriate. 

“This will be checked upon by SS and NK”. By this statement, we meant that after our findings of systematic review, the findings will be independently checked upon by two authors, namely SS and NK to examine if the findings justify the use of meta-analytic techniques. However, now that we agree that these techniques require prior writing, we removed this statement. We have provided details regarding meta-analytic techniques we intend to use in this review as follows.

Lines 252-267: “Our main outcome of interest is nonadherence to antidepressants. Non-adherence to antidepressants has the potential to be labelled as dichotomous as subjective responses (Yes/ No), or continuous (in terms of scores in use of scales). For studies with dichotomous outcomes, we plan to report odd ratios or risk ratios, while for those with continuous outcomes, such as mean scores on adherence, we will report means for individual studies first, followed by reporting a combined effect estimate. If we find sufficient studies allowing meta-analysis, we will illustrate with a forest plot to provide an overall summary of effect estimates using odd ratios and mean / standardized mean differences with confidence intervals for individual studies and overall. 

We assume that the studies included will be heterogenous, therefore, we plan to apply random effect model. We will measure the extent of heterogeneity across the studies included using I2 (Inconsistency index) (24). I2 50%-90% will be considered to have substantial heterogeneity while 75% to 100% will be considered to have considerable heterogeneity. Variance within studies will be estimated by inverse variance weighting using residual maximum likelihood (REML) technique (25).”

Lines 274-279: “We consider doing analysis of subgroups if there are good number of studies to allow subgroup analysis for example, if adequate number of studies are available in terms of specific antidepressants used to see which group of antidepressants are associated with more nonadherence to antidepressants. We also plan to do subgroup analysis based on age groups like 18 to 45, 45 to 60 and 60 and above if this is applicable.” 

Conclusions

Line 232-234 – Please be more specific in the impacts of medication adherence (i.e. remove terms like enormous suffering and add specific sequalae) and reference these statements.

Authors’ response: Thank you again for the comment. We have revised the conclusions in more specific terms as follows and referenced them.

Lines 285-288 “The consequences are increased healthcare costs due to high rates of hospitalization and emergency visits, chronicity and residual symptoms, reduced functioning, disability and poor quality of life (2-5).”

---

## [Decision Letter · Decision Letter 1]

17 Jan 2023

PONE-D-21-32408R1Protocol for the systemic review and meta-analysis of factors associated with non-adherence to antidepressants in depressive disorders in those more than 18 years of agePLOS ONE

Dear Dr. Dhungana,

Thank you for submitting your manuscript to PLOS ONE. After careful consideration, we feel that it has merit but does not fully meet PLOS ONE’s publication criteria as it currently stands. Therefore, we invite you to submit a revised version of the manuscript that addresses the points raised during the review process.

We look forward to receiving your revised manuscript.

Kind regards,

Muhammad Shahzad Aslam, Ph.D.,M.Phil., Pharm-D

Academic Editor

PLOS ONE

Journal Requirements:

Reviewers' comments:

Reviewer's Responses to Questions

**Comments to the Author**

1. Does the manuscript provide a valid rationale for the proposed study, with clearly identified and justified research questions?

Reviewer #2: Yes

2. Is the protocol technically sound and planned in a manner that will lead to a meaningful outcome and allow testing the stated hypotheses?

Reviewer #2: Yes

3. Is the methodology feasible and described in sufficient detail to allow the work to be replicable?

Reviewer #2: Yes

4. Have the authors described where all data underlying the findings will be made available when the study is complete?

Reviewer #2: Yes

5. Is the manuscript presented in an intelligible fashion and written in standard English?

Reviewer #2: Yes

6. Review Comments to the Author

You may also provide optional suggestions and comments to authors that they might find helpful in planning their study.

Reviewer #2: Thank you for addressing the comments and suggestions provided by the reviewers. The responses are comprehensive and adequately address the reviewer recommendations. I have some further very minor suggestions (line numbers refer to the tracked version of the manuscript)

Abstract

Line 43-44: please consider removing “and illustrate the results with a forest plot using Stata or Revman” as I do not think it’s necessary in the abstract. For your consideration

Methods

Line 255: Will you be using odds or risk ratios? It doesn’t matter which, but I feel that it should be decided a priori to avoid any potential bias in the analysis

Line 260: Will you be using mean differences or standardized mean differences? Again at your preference (but I would think standardized more appropriate) but should be decided a priori

7. PLOS authors have the option to publish the peer review history of their article (what does this mean?). If published, this will include your full peer review and any attached files.

Reviewer #2: No

---

## [Author Response · Author response to Decision Letter 1]

18 Jan 2023

18 January 2023

Response to reviewer

Reviewer #2: Thank you for addressing the comments and suggestions provided by the reviewers. The responses are comprehensive and adequately address the reviewer recommendations. I have some further very minor suggestions (line numbers refer to the tracked version of the manuscript)

Abstract

Line 43-44: please consider removing “and illustrate the results with a forest plot using Stata or Revman” as I do not think it’s necessary in the abstract. For your consideration

Authors’ response: Thank you for the comment. We agree with it and have removed “and illustrate the results with a forest plot using Stata or Revman” from the abstract section in lines 43-44 in tracked version.

Line 255: Will you be using odds or risk ratios? It doesn’t matter which, but I feel that it should be decided a priori to avoid any potential bias in the analysis

Line 260: Will you be using mean differences or standardized mean differences? Again at your preference (but I would think standardized more appropriate) but should be decided a priori.

Authors’ response: Thank you for this important observation. We agree it should be decided a priori to avoid potential bias in the analysis. We therefore have decided to use odd ratios and standardized mean differences for our statistical analysis purpose. Accordingly, we removed risk ratios and mean differences in lines 254 and 259 respectively in the new tracked version.

---

## [Decision Letter · Decision Letter 2]

23 Jan 2023

Protocol for the systemic review and meta-analysis of factors associated with non-adherence to antidepressants in depressive disorders in those more than 18 years of age

PONE-D-21-32408R2

Dear,

We’re pleased to inform you that your manuscript has been judged scientifically suitable for publication and will be formally accepted for publication once it meets all outstanding technical requirements.

Kind regards,

Muhammad Shahzad Aslam, Ph.D.,M.Phil., Pharm-D

Academic Editor

PLOS ONE

Additional Editor Comments (optional):

Reviewers' comments:

Reviewer's Responses to Questions

**Comments to the Author**

1. Does the manuscript provide a valid rationale for the proposed study, with clearly identified and justified research questions?

Reviewer #2: Yes

2. Is the protocol technically sound and planned in a manner that will lead to a meaningful outcome and allow testing the stated hypotheses?

Reviewer #2: Yes

3. Is the methodology feasible and described in sufficient detail to allow the work to be replicable?

Reviewer #2: Yes

4. Have the authors described where all data underlying the findings will be made available when the study is complete?

Reviewer #2: Yes

5. Is the manuscript presented in an intelligible fashion and written in standard English?

Reviewer #2: Yes

6. Review Comments to the Author

You may also provide optional suggestions and comments to authors that they might find helpful in planning their study.

Reviewer #2: Thank you for your careful revisions of this manuscript. I have no further recommendations, however there is two small typos which should be addressed prior to publication

Line 254 and 258 (tracked document): Both refer to "odd ratios" - this should be amended to "odds"

7. PLOS authors have the option to publish the peer review history of their article (what does this mean?). If published, this will include your full peer review and any attached files.

Reviewer #2: No

---

## [Editor Report · Acceptance letter]

26 Jan 2023

PONE-D-21-32408R2 

Protocol for the systematic review and meta-analysis of factors associated with non-adherence to antidepressants in depressive disorders in those more than 18 years of age 

Dear Dr. Dhungana:

I'm pleased to inform you that your manuscript has been deemed suitable for publication in PLOS ONE. Congratulations! Your manuscript is now with our production department. 

Kind regards, 

on behalf of

Dr. Muhammad Shahzad Aslam 

Academic Editor

PLOS ONE